# ALMOST TIGHT L0-NORM CERTIFIED ROBUSTNESS OF TOP-$k$ PREDICTIONS AGAINST ADVERSARIAL PERTURBATIONS

**Jinyuan Jia**
Duke University
jinyuan.jia@duke.edu

**Binghui Wang**
Illinois Institute of Technology
bwang70@iit.edu

**Xiaoyu Cao**
Duke University
xiaoyu.cao@duke.edu

**Hongbin Liu**
Duke University
hongbin.liu@duke.edu

**Neil Zhenqiang Gong**
Duke University
neil.gong@duke.edu

## ABSTRACT

Top-$k$ predictions are used in many real-world applications such as machine learning as a service, recommender systems, and web searches. $\ell_0$-norm adversarial perturbation characterizes an attack that arbitrarily modifies some features of an input such that a classifier makes an incorrect prediction for the perturbed input. $\ell_0$-norm adversarial perturbation is easy to interpret and can be implemented in the physical world. Therefore, certifying robustness of top-$k$ predictions against $\ell_0$-norm adversarial perturbation is important. However, existing studies either focused on certifying $\ell_0$-norm robustness of top-1 predictions or $\ell_2$-norm robustness of top-$k$ predictions. In this work, we aim to bridge the gap. Our approach is based on randomized smoothing, which builds a provably robust classifier from an arbitrary classifier via randomizing an input. Our major theoretical contribution is an almost tight $\ell_0$-norm certified robustness guarantee for top-$k$ predictions. We empirically evaluate our method on CIFAR10 and ImageNet. For instance, our method can build a classifier that achieves a certified top-3 accuracy of 69.2% on ImageNet when an attacker can arbitrarily perturb 5 pixels of a testing image.

## 1 INTRODUCTION

Adversarial example is a well-known severe security vulnerability of classifiers. Specifically, given a classifier $f$ and a testing input $\mathbf{x}$, an attacker can carefully craft a human-imperceptible perturbation $\delta$ such that $f(x) \neq f(x + \delta)$. The perturbation $\delta$ is called *adversarial perturbation*, while the input $x + \delta$ is called an *adversarial example*. Many *empirical* defenses (Goodfellow et al., 2015; Na et al., 2018; Metzen et al., 2017; Svoboda et al., 2019; Buckman et al., 2018; Ma et al., 2018; Guo et al., 2018; Dhillon et al., 2018; Xie et al., 2018; Song et al., 2018) have been developed to defend against adversarial examples in the past several years. However, these empirical defenses were often soon broken by strong adaptive adversaries (Carlini & Wagner, 2017; Athalye et al., 2018; Uesato et al., 2018; Athalye & Carlini, 2018). To end this cat-and-mouse game, many *certified* defenses (Scheibler et al., 2015; Carlini et al., 2017; Ehlers, 2017; Katz et al., 2017; Cheng et al., 2017; Lomuscio & Maganti, 2017; Fischetti & Jo, 2018; Bunel et al., 2018; Wong & Kolter, 2018; Wong et al., 2018; Raghunathan et al., 2018a;b; Dvijotham et al., 2018a;b; Gehr et al., 2018; Mirman et al., 2018; Singh et al., 2018; Weng et al., 2018; Zhang et al., 2018; Gowal et al., 2018; Wang et al., 2018; Lecuyer et al., 2019; Li et al., 2019; Cohen et al., 2019; Lee et al., 2019; Salman et al., 2019; Wang et al., 2020; Jia et al., 2020; Zhai et al., 2020) have been proposed. In particular, a classifier $f$ is said to be certifiably robust for an input $\mathbf{x}$ if it provably predicts the same top-1 label (i.e., $f(x) = f(x + \delta)$) when the adversarial perturbation $\delta$ is bounded, e.g., the $\ell_p$-norm of $\delta$ is smaller than a threshold. The threshold is also called *certified radius*. In this work, we focus on $\ell_0$-norm adversarial perturbation, which arbitrarily manipulates some features of a testing input and can be implemented in the physical world.

However, most existing certified defenses focus on top-1 predictions. In many applications, top-$k$ predictions that return the $k$ most likely labels are more relevant. For instance, when a classifier is deployed as a cloud service (also called *machine learning as a service*) (Google Cloud Vision; Microsoft; Amazon AWS; Clarifai), top-$k$ labels for a testing input are often returned to a customer for more informed decisions; in recommender systems and web searches, top-$k$ items/webpages are recommended to a user. Despite the importance and relevance of top-$k$ predictions, their certified robustness against adversarial perturbations is largely unexplored. One exception is the recent work from Jia et al. (2020), which derived a tight $\ell_2$-norm certified robustness for top-$k$ predictions. Such $\ell_2$-norm certified robustness can be transformed to $\ell_0$-norm certified robustness via employing the inequality between $\ell_0$-norm and $\ell_2$-norm. However, the $\ell_0$-norm certified robustness derived from such transformations is suboptimal.

**Our work:** We aim to develop $\ell_0$-norm certified robustness of top-$k$ predictions. Our approach is based on *randomized smoothing* (Cao & Gong, 2017; Liu et al., 2018; Lecuyer et al., 2019; Li et al., 2019; Cohen et al., 2019; Lee et al., 2019; Jia et al., 2020; Levine & Feizi, 2019), which can build a certifiably robust classifier from any base classifier via randomizing the input. We adopt randomized smoothing because it is applicable to any classifier and scalable to large neural networks. In particular, we use a randomized smoothing method called *randomized ablation* (Levine & Feizi, 2019), which achieves state-of-the-art $\ell_0$-norm certified robustness for top-1 predictions. Unlike other randomized smoothing methods (Cao & Gong, 2017; Lecuyer et al., 2019; Li et al., 2019; Cohen et al., 2019) that randomize an input via adding *additive* noise (e.g., Gaussian, Laplacian, or discrete noise) to it, randomized ablation randomizes an input via subsampling its features. Specifically, given an arbitrary classifier (called *base classifier*) and a testing input $\mathbf{x}$, randomized ablation creates an *ablated input* via retaining some randomly selected features in $\mathbf{x}$ and setting the remaining features to a special value, e.g., median of the feature value, mean of the feature value, or a special symbol. When the testing input is an image, the features are the image's pixels. Then, we feed the ablated input to the base classifier. Since the ablated input is random, the output of the base classifier is also random. Specifically, we denote by $p_j$ the probability that the base classifier outputs a label $j$ for the random ablated input. The original randomized ablation method builds a *smoothed classifier* that outputs the label with the largest label probability $p_j$ for a testing input $\mathbf{x}$. In our work, the smoothed classifier returns the $k$ labels with the largest label probabilities for $\mathbf{x}$.

Our major theoretical contribution is an almost tight $\ell_0$-norm certified robustness guarantee of top-$k$ predictions for the smoothed classifier constructed by randomized ablation. Specifically, we first derive an $\ell_0$-norm certified robustness guarantee of top-$k$ predictions for the smoothed classifier. Our results show that a label $l$ is provably among the top-$k$ labels predicted by the smoothed classifier for a testing input $\mathbf{x}$ when the attacker arbitrarily perturbs at most $r_l$ features of $\mathbf{x}$, where $r_l$ is the $\ell_0$-norm certified radius. Moreover, we prove that our certified radius is *tight* when $k = 1$ and is *almost tight* when $k > 1$. In particular, if no assumptions on the base classifier are made, it is impossible to derive a certified radius that is larger than $r_l + \mathbb{I}(k \neq 1)$. In other words, when an attacker manipulates at least $r_l + 1 + \mathbb{I}(k \neq 1)$ features of a testing input, there exists a base classifier from which the smoothed classifier's top-$k$ predicted labels do not include $l$ or there exist ties.

Our work has several technical differences with Levine & Feizi (2019). First, we derive the $\ell_0$-norm certified radius of top-$k$ predictions for randomized ablation, while Levine & Feizi (2019) only derived the certified radius of top-1 predictions. Second, our certified radius is the same as or larger than that in Levine & Feizi (2019) for top-1 predictions, because we leverage the discrete property of the label probabilities to derive our certified radius. Third, we prove the (almost) tightness of the certified radius, while Levine & Feizi (2019) didn't. Our work also has several technical differences with Jia et al. (2020), which derived a tight $\ell_2$-norm certified radius of top-$k$ predictions for randomized smoothing with Gaussian additive noise. Since they add additive Gaussian noise to a testing input, the space of randomized inputs is continuous. However, our space of ablated inputs is discrete, as we randomize a testing input via subsampling its features. As a result, Jia et al. and our work use substantially different techniques to derive the $\ell_2/\ell_0$-norm certified radiuses and prove their (almost) tightness. In particular, when deriving the $\ell_2/\ell_0$-norm certified radiuses, our work needs to construct different regions in the discrete space of ablated inputs such that the Neyman-Pearson Lemma (Neyman & Pearson, 1933) can be applied. When proving the (almost) tightness, we use a completely different approach from Jia et al.. First, Jia et al. relies on the Intermediate Value Theorem, which is not applicable to our discrete data. Second, since Gaussian noise is not uniform, Jia et al. need to prove the results via Mathematical Induction. However, Mathematical Induction is

unnecessary in our case because the ablated inputs that can be derived from an input are uniformly distributed in the space of ablated inputs.

We evaluate our method on CIFAR10 and ImageNet. Our results show that our method substantially outperforms state-of-the-art for top-$k$ predictions. For instance, our method achieves a certified top-3 accuracy of 69.2% on ImageNet when an attacker arbitrarily perturbs 5 pixels of a testing image. Under the same setting, Jia et al. (2020) achieves a certified top-3 accuracy of only 9.0%, when transforming their $\ell_2$-norm certified robustness to $\ell_0$-norm certified robustness.

Our contributions can be summarized as follows:

- We derive an $\ell_0$-norm certified radius of top-$k$ predictions for randomized ablation.
- We prove that our certified radius is tight when $k = 1$ and almost tight when $k > 1$.
- We empirically evaluate our method on CIFAR10 and ImageNet.

## 2 THEORETICAL RESULTS

In this section, we show our core theoretical contributions.

### 2.1 BUILDING A SMOOTHED CLASSIFIER VIA RANDOMIZED ABLATION

Suppose we have a base classifier $f$, which classifies a testing input $\mathbf{x}$ to one of $c$ classes $\{1, 2, \cdots, c\}$ deterministically. For simplicity, we assume $\mathbf{x}$ is an image with $d$ pixels. Given an input $\mathbf{x}$, randomized ablation (Levine & Feizi, 2019) creates an *ablated input* as follows: we first randomly subsample $e$ pixels from $\mathbf{x}$ without replacement and keep their values. Then, we set the remaining pixel values in the ablated input to a special value, e.g., median of the pixel value, mean of the pixel value, or a special symbol. When the image is a color image, we set the values of the three channels of each pixel separately. Note that an ablated input has the same size with $\mathbf{x}$. We use $h(\mathbf{x}, e)$ to denote the randomly ablated input for simplicity. Given $h(\mathbf{x}, e)$ as input, the output of the base classifier $f$ is also random. We use $p_j$ to denote the probability that the base classifier $f$ predicts class $j$ when taking $h(\mathbf{x}, e)$ as input, i.e., $p_j = \Pr(f(h(\mathbf{x}, e)) = j)$. Note that $p_j$ is an integer multiple of $\frac{1}{\binom{d}{e}}$, which we will leverage to derive a tighter certified robustness guarantee. We build a smoothed classifier $g$ that outputs the $k$ labels with the largest label probabilities $p_j$'s for $\mathbf{x}$. Moreover, we denote by $g_k(\mathbf{x})$ the set of $k$ labels predicted for $\mathbf{x}$.

### 2.2 DERIVING THE CERTIFIED RADIUS FOR THE SMOOTHED CLASSIFIER

**Defining two random variables:** Suppose an attacker adds a perturbation $\delta$ to an input $\mathbf{x}$, where $\|\delta\|_0$ is the number of pixels perturbed by the attacker. We define the following two random variables:

$$U = h(\mathbf{x}, e), V = h(\mathbf{x} + \delta, e), \tag{1}$$

where the random variables $U$ and $V$ denote the ablated inputs derived from $\mathbf{x}$ and its perturbed version $\mathbf{x} + \delta$, respectively. We use $\mathcal{S}$ to denote the joint space of $U$ and $V$, i.e., $\mathcal{S}$ is the set of ablated inputs that can be derived from $\mathbf{x}$ or $\mathbf{x} + \delta$. Given the definition of $U$ and $V$, $\Pr(f(U) = j)$ and $\Pr(f(V) = j)$ respectively represent the label probabilities of the input $\mathbf{x}$ and its perturbed version $\mathbf{x} + \delta$ predicted by the smoothed classifier.

**Derivation goal:** Intuitively, an ablated input $h(\mathbf{x}, e)$ is very likely to not include any perturbed pixel if $\|\delta\|_0$ is bounded and $e$ is relatively small, and thus the predicted labels of the smoothed classifier are not influenced by the perturbation. Formally, our goal is to show that a label $l \in \{1, 2, \cdots, c\}$ is provably in the top-$k$ labels predicted by the smoothed classifier for an input $\mathbf{x}$ when the number of perturbed pixels is no larger than a threshold. In other words, we aim to show that $l \in g_k(\mathbf{x} + \delta)$ when $\|\delta\|_0 \leq r_l$, where $r_l$ is the certified radius. Our key idea to derive the certified radius is to guarantee that, when taking $V$ as input, the label probability for label $l$ is larger than the smallest one among the label probabilities of any $k$ labels from all labels except $l$. We let $\Gamma = \{1, 2, \cdots, c\} \setminus \{l\}$, i.e., $\Gamma$ denotes the set of all labels except $l$. We use $\Gamma_k$ to denote a set of $k$ labels in $\Gamma$. Then, we aim to find a maximum certified radius $r_l$ such that:

$$\Pr(f(V) = l) > \max_{\Gamma_k \subset \Gamma} \min_{j \in \Gamma_k} \Pr(f(V) = j). \tag{2}$$

Roughly speaking, the above equation means that: to ensure $l$ exists in the set of top-$k$ labels, $\Pr(f(V) = l)$ should be larger than the minimum probability observed by taking any set of $k$ labels $\Gamma_k$ excluding $l$. To reach the goal, we derive a lower bound of $\Pr(f(V) = l)$ and an upper bound of $\max_{\Gamma_k \subset \Gamma} \min_{j \in \Gamma_k} \Pr(f(V) = j)$. In particular, we derive a lower bound and an upper bound using the probabilities that $V$ is in certain regions of the discrete space $\mathcal{S}$, and such probabilities can be efficiently computed for $\forall \|\delta\|_0 = r$. Then, we can leverage binary search to find the maximum $r$ such that the lower bound is larger than the upper bound and treat the maximum $r$ as the certified radius $r_l$. Next, we respectively introduce how to derive lower and upper bounds and use them to compute certified radius.

**Deriving a lower bound of $\mathbf{Pr}(f(V) = l)$ and an upper bound of** $\max_{\Gamma_k \subset \Gamma} \min_{j \in \Gamma_k} \mathbf{Pr}(f(V) = j)$**:** We show our intuition to derive the upper and lower bounds. Our formal analysis is shown in the proof of Theorem 1. Our idea to derive the bounds is to divide the discrete space $\mathcal{S}$ in an innovative way such that we can leverage the Neyman-Pearson Lemma (Neyman & Pearson, 1933). Suppose for the random variable $U$, we have a lower bound of the label probability for $l$ and an upper bound of the label probability for every other label. Formally, we have $\underline{p}_l, \overline{p}_1 \cdots \overline{p}_{l-1}, \overline{p}_l, \cdots, \overline{p}_c$ that satisfy the following:

$$\underline{p}_l \le \Pr(f(U) = l), \overline{p}_j \ge \Pr(f(U) = j), \forall j \ne l, \tag{3}$$

where $\underline{p}$ and $\overline{p}$ denote the lower and upper bounds of $p$, respectively. Multiplying each term in Equation (3) by $\binom{d}{e}$, we have $\underline{p}_l \cdot \binom{d}{e} \le \Pr(f(U) = l) \cdot \binom{d}{e}, \overline{p}_j \cdot \binom{d}{e} \ge \Pr(f(U) = j) \cdot \binom{d}{e}, \forall j \ne l$. Since $p_l$ and $p_j (\forall j \ne l)$ are integer multiples of $\frac{1}{\binom{d}{e}}$, we have $\lceil \underline{p}_l \cdot \binom{d}{e} \rceil \le \Pr(f(U) = l) \cdot \binom{d}{e}, \lfloor \overline{p}_j \cdot \binom{d}{e} \rfloor \ge \Pr(f(U) = j) \cdot \binom{d}{e}, \forall j \ne l$. Therefore, we have the following:

$$\underline{p}'_l \triangleq \frac{\lceil \underline{p}_l \cdot \binom{d}{e} \rceil}{\binom{d}{e}} \le \Pr(f(U) = l), \overline{p}'_j \triangleq \frac{\lfloor \overline{p}_j \cdot \binom{d}{e} \rfloor}{\binom{d}{e}} \ge \Pr(f(U) = j), \forall j \ne l. \tag{4}$$

Let $\overline{p}_{a_k} \ge \overline{p}_{a_{k-1}} \cdots \ge \overline{p}_{a_1}$ be the $k$ largest ones among $\{\overline{p}_1, \cdots, \overline{p}_{l-1}, \overline{p}_{l+1}, \cdots, \overline{p}_c\}$, where ties are broken uniformly at random. We denote $\Upsilon_t = \{a_1, a_2, \cdots, a_t\}$ as the set of $t$ labels with the smallest label probability upper bounds in the $k$ largest ones and denote by $\overline{p}'_{\Upsilon_t} = \sum_{j \in \Upsilon_t} \overline{p}'_j$ the sum of the $t$ label probability bounds, where $t = 1, 2, \cdots, k$.

We define regions $\mathcal{A}$, $\mathcal{B}$, and $\mathcal{C}$ in $\mathcal{S}$ as the sets of ablated inputs that can be derived only from $\mathbf{x}$, only from $\mathbf{x} + \delta$, and from both $\mathbf{x}$ and $\mathbf{x} + \delta$, respectively. In particular, the region $\mathcal{A}$ is a set of ablated inputs that can only be obtained via sampling $e$ features (pixels) from $\mathbf{x}$. The region $\mathcal{B}$ is a set of ablated inputs that can only be obtained via sampling $e$ features from the perturbed $\mathbf{x} + \delta$, and the region $\mathcal{C}$ is a set of ablated inputs that can be obtained via sampling $e$ features from both $\mathbf{x}$ and $\mathbf{x} + \delta$. Then, we can find a region $\mathcal{A}' \subseteq \mathcal{C}$ such that $\Pr(U \in \mathcal{A}' \cup \mathcal{A}) = \underline{p}'_l$. Note that we assume we can find such a region $\mathcal{A}'$ since we aim to find sufficient condition. Similarly, we can find $\mathcal{H}_{\Upsilon_t} \in \mathcal{C}$ such that we have $\Pr(U \in \mathcal{H}_{\Upsilon_t}) = \overline{p}'_{\Upsilon_t}$. Then, we can apply the Neyman-Pearson Lemma (Neyman & Pearson, 1933) to derive a lower bound of $\Pr(f(V) = l)$ and an upper bound of $\max_{\Gamma_k \subset \Gamma} \min_{j \in \Gamma_k} \Pr(f(V) = j)$ by leveraging the probabilities of $V$ in regions $\mathcal{A}' \cup \mathcal{A}$ and $\mathcal{H}_{\Upsilon_t} \cup \mathcal{B}$. Formally, we have the following:

$$\Pr(f(V) = l) \ge \Pr(V \in \mathcal{A}' \cup \mathcal{A}), \max_{\Gamma_k \subset \Gamma} \min_{j \in \Gamma_k} \Pr(f(V) = j) \le \min_{t=1}^{k} \frac{\Pr(V \in \mathcal{H}_{\Upsilon_t} \cup \mathcal{B})}{t}. \tag{5}$$

**Computing certified radius:** Given the lower and upper bounds, we can find the maximum $r = \|\delta\|_0$ such that the lower bound $\Pr(V \in \mathcal{A}' \cup \mathcal{A})$ is larger than the upper bound $\min_{t=1}^{k} \frac{\Pr(V \in \mathcal{H}_{\Upsilon_t} \cup \mathcal{B})}{t}$. The maximum $r$ is the certified radius. Formally, we have the following theorem:

**Theorem 1** ($\ell_0$-norm Certified Radius for Top-$k$ Predictions)**.** *Suppose we have an input $\mathbf{x}$ with $d$ features, a deterministic base classifier $f$, an integer $e$, a smoothed classifier $g$ where $g_k(\mathbf{x})$ is a set of $k$ labels predicted for $\mathbf{x}$, an arbitrary label $l \in \{1, 2, \cdots, c\}$, $\underline{p}_l, \overline{p}_1, \cdots, \overline{p}_{l-1}, \overline{p}_{l+1}, \cdots, \overline{p}_c$ that satisfy Equation (3), and $\underline{p}'_l, \overline{p}'_j (\forall j \ne l)$ that are defined in Equation (4). If the following optimization problem has a solution $r_l$:*

$$r_l = \arg\max_{r \ge 0} r \quad s.t. \ \underline{p}'_l - (1 - \frac{\binom{d-r}{e}}{\binom{d}{e}}) > \min_{t=1}^{k} \frac{\overline{p}'_{\Upsilon_t} + (1 - \frac{\binom{d-r}{e}}{\binom{d}{e}})}{t}. \tag{6}$$

*Then, we have the following:*

$$l \in g_k(\mathbf{x} + \delta), \forall \|\delta\|_0 \leq r_l. \tag{7}$$

*Proof.* Please refer to Appendix A. □

Next, we show that our derived certified radius is (almost) tight. In particular, when using randomized ablation and no further assumptions are made on the base classifier, it is impossible to certify an $\ell_0$-norm radius that is larger than $r_l + \mathbb{I}(k \neq 1)$ for top-$k$ predictions.

**Theorem 2** (Almost Tightness of our Certified Radius). *Assuming we have $\binom{d-r_l-2}{e-1} \geq 1$, $\underline{p'_l} + \sum_{j \in \Upsilon_k} \overline{p'_j} \leq 1$, and $\underline{p'_l} + \sum_{j \neq l} \overline{p'_j} \geq 1$. If no assumption on the base classifier is made, then, for any perturbation $\|\delta\|_0 > r_l + \mathbb{I}(k \neq 1)$, there exists a base classifier $f^*$ consistent with Equation (3) but we have $l \notin g_k(\mathbf{x} + \delta)$ or there exist ties.*

*Proof.* Please refer to Appendix B. □

**Comparing with Levine & Feizi (2019) when $k = 1$:** Our certified radius reduces to the maximum $r$ that satisfies $\underline{p'_l} - \overline{p'_{a_1}} > 2 \cdot (1 - \frac{\binom{d-r}{e}}{\binom{d}{e}})$ when $k = 1$. In contrast, the certified radius in Levine & Feizi (2019) is the maximum $r$ that satisfies $\underline{p_l} - \overline{p_{a_1}} > 2 \cdot (1 - \frac{\binom{d-r}{e}}{\binom{d}{e}})$. Since $\underline{p'_l} \geq \underline{p_l}$ and $\overline{p'_{a_k}} \leq \overline{p_{a_k}}$, our certified radius is the same as or larger than that in Levine & Feizi (2019). Note that the method by Levine & Feizi (2019) cannot be extended in a straightforward way to derive the certified robustness for top-$k$ predictions. The reason is that they consider each label independently in their derivation. In particular, they use the probability that the base classifier predicts label $i$ for an ablated clean input (i.e., $\Pr(f(U) = i)$) to bound the probability that the base classifier predicts label $i$ for an ablated adversarial input (i.e., $\Pr(f(V) = i)$). However, to derive the certified robustness for top-$k$ predictions, we need to jointly derive the bounds for multiple label probabilities (i.e., Equation (5)). Moreover, because of the difference in deriving label-probability bounds, our techniques are significantly different. In particular, Levine & Feizi only need the law of total probability while we leverage Neyman-Pearson Lemma. Moreover, Levine & Feizi (2019) did not analyze the tightness of the certified radius for top-1 predictions.

**Comparing with Jia et al. (2020):** Jia et al. (2020) proved the exact tightness of their $\ell_2$-norm certified radius for randomized smoothing with Gaussian noise. We highlight that our techniques to prove our almost tightness are substantially different from those in Jia et al.. First, they proved the existence of a region via the Intermediate Value Theorem, which relies on the continuity of Gaussian noise. However, our space of ablated inputs is discrete. Therefore, given a probability upper/lower bound, it is challenging to find a region whose probability measure exactly equals to the given value, since the Intermediate Value Theorem is not applicable. As a result, we cannot prove the exact tightness of the $\ell_0$-norm certified radius when $k > 1$. To address the challenge, we find a region whose probability measure is slightly smaller than the given upper bound, which enables us to prove the almost tightness of our certified radius. Second, since Gaussian noise is not uniform, they need to iteratively construct regions via leveraging Mathematical Induction. However, Mathematical Induction is unnecessary in our case because the ablated inputs that can be derived from an input are uniformly distributed in the space of ablated inputs.

**Computing $r_l$ in practice:** When applying our Theorem 1 to calculate the certified radius $r_l$ in practice, we need the probability bounds $\underline{p'_l}$ and $\overline{p'_{\Upsilon_t}}$ and solve the optimization problem in Equation (6). We can leverage a Monte Carlo method developed by Jia et al. (2020) to estimate the probability bounds ($\underline{p_l}$ and $\overline{p_j}, \forall j \neq l$) with probabilistic guarantees. Then, we can use them to estimate $\underline{p'_l}$ and $\overline{p'_{\Upsilon_t}}$. Moreover, given the probability bounds $\underline{p'_l}$ and $\overline{p'_{\Upsilon_t}}$, we can use binary search to solve Equation (6) to find the certified radius $r_l$.

Specifically, the probabilities $p_1, p_2, \cdots, p_c$ can be viewed as a multinomial distribution over the label set $\{1, 2, \cdots, c\}$. Given $h(\mathbf{x}, e)$ as input, $f(h(\mathbf{x}, e))$ can be viewed as a sample from the multinomial distribution. Therefore, estimating $\underline{p_l}$ and $\overline{p_i}$ for $i \neq l$ is essentially a one-sided *simultaneous confidence interval* estimation problem. In particular, we leverage the simultaneous

confidence interval estimation method called SimuEM (Jia et al., 2020) to estimate these bounds with a confidence level at least $1 - \alpha$. Specifically, given an input $\mathbf{x}$ and parameter $e$, we randomly create $n$ ablated inputs, i.e., $\epsilon^1, \epsilon^2, \cdots, \epsilon^n$. We denote by $n_j$ the frequency of the label $j$ predicted by the base classifier for the $n$ ablated inputs. Formally, we have $n_j = \sum_{i=1}^{n} \mathbb{I}(f(\epsilon^i) = j)$, where $j \in \{1, 2, \cdots, c\}$ and $\mathbb{I}$ is the indicator function. According to Jia et al. (2020), we have the following probability bounds with a confidence level at least $1 - \alpha$:

$$\underline{p_l} = B\left(\frac{\alpha}{c}; n_l, n - n_l + 1\right), \overline{p}_j = B(1 - \frac{\alpha}{c}; n_j + 1, n - n_j), \ \forall j \neq l, \tag{8}$$

where $B(q; \xi, \zeta)$ is the $q$th quantile of a beta distribution with shape parameters $\xi$ and $\zeta$. Then, we can compute $\underline{p}'_l$ and $\overline{p}'_j, \forall j \neq l$ based on Equation (4). Finally, we estimate $\overline{p}'_{\Upsilon_t}$ as $\overline{p}'_{\Upsilon_t} = \min(\sum_{j \in \Upsilon_t} \overline{p}'_j, 1 - \underline{p}'_l)$.

# 3 EVALUATION

## 3.1 EXPERIMENTAL SETUP

**Datasets and models:** We use CIFAR10 (Krizhevsky et al., 2009) and ImageNet (Deng et al., 2009) for evaluation. We normalize pixel values to be in the range [0,1]. We use the publicly available implementation[1] of randomized ablation to train our models. In particular, we use ResNet-110 and RestNet-50 as the base classifiers for CIFAR10 and ImageNet, respectively. Moreover, as in Lee et al. (2019), we use 500 testing examples for both CIFAR10 and ImageNet.

**Parameter setting:** Unless otherwise mentioned, we adopt the following default parameters. We set $e = 50$ and $e = 1,000$ for CIFAR10 and ImageNet, respectively. We set $k = 3$, $n = 100,000$, and $\alpha = 0.001$. We will study the impact of each parameter while fixing the remaining ones to their default values.

**Evaluation metric:** We use the *certified top-$k$ accuracy* as an evaluation metric. Specifically, given a number of perturbed pixels, certified top-$k$ accuracy is the fraction of testing images, whose true labels have $\ell_0$-norm certified radiuses for top-$k$ predictions that are no smaller than the given number of perturbed pixels. Note that our $\ell_0$-norm certified radius corresponds to the maximum number of pixels that can be perturbed by an attacker.

**Compared methods:** We compare six randomized smoothing based methods. The first four are only applicable for top-1 predictions, while the latter two are applicable for top-$k$ predictions.

- **Cohen et al. (2019).** This method adds Gaussian noise to a testing image and derives a tight $\ell_2$-norm certified radius for top-1 predictions. In particular, considering the three color channels and each pixel value is normalized to be in the range [0,1], an $\ell_0$-norm certified number of perturbed pixels $r_l$ can be obtained from an $\ell_2$-norm certified radius $\sqrt{3r_l}$.

- **Lee et al. (2019).** This method derives an $\ell_0$-norm certified radius for top-1 predictions. This method is applicable to discrete features. Like Lee et al. (2019), we treat the pixel values as discrete in the domain $\{0, 1/256, 2/256, \cdots, 255/256\}$. Since their $\ell_0$-norm certified radius is for pixel channels (each pixel has 3 color channels), a certified number of perturbed pixels $r_l$ can be obtained from their $\ell_0$-norm certified radius $3r_l$.

- **Levine & Feizi (2019).** This method derives an $\ell_0$-norm certified number of perturbed pixels for top-1 predictions in randomized ablation. This method requires a lower bound of the largest label probability and an upper bound of the second largest label probability to calculate the certified number of perturbed pixels. They estimated the lower bound using the Monte Carlo method in Cohen et al. (2019) and the upper bound as 1 - the lower bound. Note that our certified radius is theoretically no smaller than that in Levine & Feizi (2019) when $k = 1$. Therefore, we use our derived certified radius when evaluating this method. We also found that the top-1 certified accuracies based on our derived certified radius and their derived certified radius have negligible differences on CIFAR10 and ImageNet, and thus we do not show the differences for simplicity.

---

[1]https://github.com/alevine0/randomizedAblation/

**Table 1: Certified top-$k$ accuracies of the compared methods on CIFAR10.**

| #Perturbed pixels | | 1 | 2 | 3 | 4 | 5 |
|---|---|---|---|---|---|---|
| Certified top-1 accuracy | Cohen et al. (2019) | 0.118 | 0.056 | 0.018 | 0.0 | 0.0 |
| | Lee et al. (2019) | 0.188 | 0.018 | 0.004 | 0.002 | 0.0 |
| | Levine & Feizi (2019) | 0.704 | 0.680 | 0.670 | 0.646 | 0.610 |
| | **Levine & Feizi (2019) + SimuEM (Jia et al., 2020)** | **0.746** | **0.718** | **0.690** | **0.660** | **0.636** |
| | **Our method** | **0.746** | **0.718** | **0.690** | **0.660** | **0.636** |
| Certified top-3 accuracy | Jia et al. (2020) | 0.244 | 0.124 | 0.070 | 0.028 | 0.004 |
| | **Our method** | **0.886** | **0.860** | **0.838** | **0.814** | **0.780** |

**Table 2: Certified top-$k$ accuracies of the compared methods on ImageNet.**

| #Perturbed pixels | | 1 | 2 | 3 | 4 | 5 |
|---|---|---|---|---|---|---|
| Certified top-1 accuracy | Cohen et al. (2019) | 0.226 | 0.152 | 0.120 | 0.088 | 0.0 |
| | Lee et al. (2019) | 0.338 | 0.196 | 0.104 | 0.092 | 0.070 |
| | Levine & Feizi (2019) | 0.602 | 0.600 | 0.596 | 0.588 | 0.586 |
| | **Levine & Feizi (2019) + SimuEM (Jia et al., 2020)** | **0.634** | **0.628** | **0.618** | **0.616** | **0.608** |
| | **Our method** | **0.634** | **0.628** | **0.618** | **0.616** | **0.608** |
| Certified top-3 accuracy | Jia et al. (2020) | 0.326 | 0.232 | 0.160 | 0.120 | 0.090 |
| | **Our method** | **0.740** | **0.730** | **0.712** | **0.698** | **0.692** |

- **Levine & Feizi (2019) + SimuEM (Jia et al., 2020)**. This is the Levine & Feizi (2019) method with the lower/upper bounds of label probabilities estimated using the simultaneous confidence interval estimation method called SimuEM. Again, we use our derived certified radius for top-1 predictions in this method.

- **Jia et al. (2020)**. This work extends Cohen et al. (2019) from top-1 predictions to top-$k$ predictions. In detail, they derive a tight $\ell_2$-norm certified radius of top-$k$ predictions for randomized smoothing with Gaussian noise. An $\ell_0$-norm certified number of perturbed pixels $r_l$ for top-$k$ predictions can be obtained from an $\ell_2$-norm certified radius $\sqrt{3r_l}$.

- **Our method**. Our method produces an almost tight $\ell_0$-norm certified number of perturbed pixels of top-$k$ predictions.

Note that we compare with Cohen et al. (2019) and Jia et al. (2020) because we aim to show that it is suboptimal to derive $\ell_0$-norm certified robustness for top-$k$ predictions by leveraging the relationship between $\ell_2$-norm and $\ell_0$-norm.

## 3.2 Experimental Results

**Comparison results:** Table 1 and 2 respectively show the certified top-$k$ accuracies of the compared methods on CIFAR10 and ImageNet when an attacker perturbs a certain number of pixels. The Gaussian noise in Cohen et al. (2019) and Jia et al. (2020) has mean 0 and standard deviation $\sigma$. We obtain the certified top-$k$ accuracies for different $\sigma$, i.e., we explored $\sigma = 0.1, 0.12, 0.25, 0.5, 1.0$. Lee et al. (2019) has a noise parameter $\beta$. We obtain the certified top-1 accuracies for different $\beta$. In particular, we explored $\beta = 0.1, 0.2, 0.3, 0.4, 0.5$, which were also used by Lee et al. (2019). Then, we report the largest certified top-$k$ accuracies of Cohen et al. (2019), Lee et al. (2019), and Jia et al. (2020) for each given number of perturbed pixels. We use the default values of $e$ for Levine & Feizi (2019) and our method.

We have two observations from Table 1 and 2. First, our method substantially outperforms Jia et al. (2020) for top-$k$ predictions, while Levine & Feizi (2019) substantially outperforms Cohen et al. (2019) and Lee et al. (2019) for top-1 predictions. Since our method and Levine & Feizi (2019) use randomized ablation, while the remaining methods use additive noise (Gaussian or discrete noise) to randomize a testing input, our results indicate that randomized ablation is superior to additive noise at certifying $\ell_0$-norm robustness. Second, Levine & Feizi (2019) + SimuEM (Jia et al., 2020)

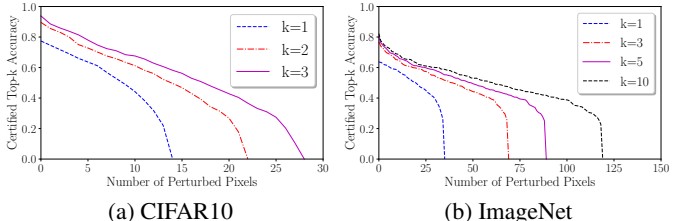

(a) CIFAR10          (b) ImageNet

**Figure 1: Impact of $k$ on certified top-$k$ accuracy.**

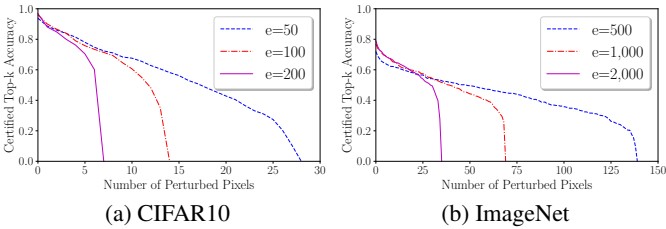

(a) CIFAR10          (b) ImageNet

**Figure 2: Impact of $e$ on certified top-$3$ accuracy.**

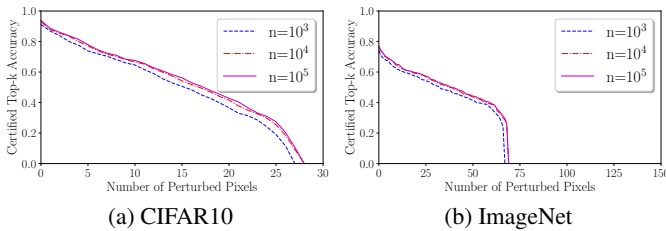

(a) CIFAR10          (b) ImageNet

**Figure 3: Impact of $n$ on certified top-$3$ accuracy.**

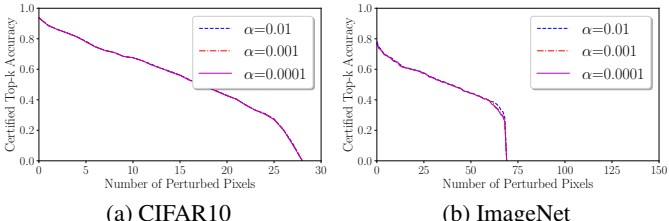

(a) CIFAR10          (b) ImageNet

**Figure 4: Impact of $\alpha$ on certified top-$3$ accuracy.**

outperforms Levine & Feizi (2019). This is because SimuEM can more accurately estimate the label probability bounds via simultaneous confidence interval estimations.

**Impact of $k$, $e$, $n$, and $\alpha$:** Figure 1, 2, 3 and 4 show the certified top-$k$ accuracy of our method vs. number of perturbed pixels for different $k$, $e$, $n$, and $\alpha$, respectively. Naturally, the certified top-$k$ accuracy increases as $k$ increases. For instance, when 5 pixels are perturbed, the certified top-1 and top-3 accuracies are 63.6% and 78.0% on CIFAR10, respectively. We observe that $e$ provides a tradeoff between accuracy under no attacks and robustness. Specifically, when $e$ is larger, the accuracy under no attacks (i.e., certified accuracy with 0 perturbed pixels) is higher, while the certified accuracy decreases to 0 more quickly as the number of perturbed pixels increases. As $n$ becomes larger, the curve of the certified accuracy may become higher. The reason is that a larger $n$ makes the estimated label probability bounds $\underline{p_l}$ and $\overline{p}_{\Upsilon_t}$ tighter and thus the $\ell_0$-norm certified radius may be larger, which result in a larger certified accuracy. Theoretically, as the confidence level $1 - \alpha$ decreases, the curve of the certified accuracy may become higher. This is because a smaller confidence level leads to tighter estimated label probability bounds $\underline{p_l}$ and $\overline{p}_{\Upsilon_t}$, and thus the certified accuracy may be larger. However, we observe the differences between different confidence levels are negligible when the confidence levels are high enough (i.e., $\alpha$ is small enough).

## 4 RELATED WORK

Many certified defenses have been proposed to defend against adversarial perturbations. These defenses leverage various techniques including satisfiability modulo theories (Scheibler et al., 2015; Carlini et al., 2017; Ehlers, 2017; Katz et al., 2017), interval analysis (Wang et al., 2018), linear programming (Cheng et al., 2017; Lomuscio & Maganti, 2017; Fischetti & Jo, 2018; Bunel et al., 2018; Wong & Kolter, 2018; Wong et al., 2018), semidefinite programming (Raghunathan et al., 2018a;b), dual optimization (Dvijotham et al., 2018a;b), abstract interpretation (Gehr et al., 2018; Mirman et al., 2018; Singh et al., 2018), and layer-wise relaxation (Weng et al., 2018; Zhang et al., 2018; Gowal et al., 2018; Chiang et al., 2020). However, these defenses suffer from one or two limitations: 1) they are not scalable to large neural networks and/or 2) they are only applicable to specific neural network architectures. Randomized smoothing addresses the two limitations. Next, we review randomized smoothing based methods for certifying non-$\ell_0$-norm and $\ell_0$-norm robustness.

**Randomized smoothing for non-$\ell_0$-norm robustness:** Randomized smoothing was first proposed as an empirical defense (Cao & Gong, 2017; Liu et al., 2018). In particular, Cao & Gong (2017) proposed to use uniform random noise from a hypercube centered at a testing example to smooth its predicted label. Lee et al. (2019) derived certified robustness for such uniform random noise. Lecuyer et al. (2019) was the first to derive formal $\ell_2$ and $\ell_\infty$-norm robustness guarantee of randomized smoothing with Gaussian or Laplacian noise via differential privacy techniques. Subsequently, Li et al. (2019) leveraged information theory to derive a tighter $\ell_2$-norm robustness guarantee. Cohen et al. (2019) leveraged the Neyman-Pearson Lemma (Neyman & Pearson, 1933) to obtain a tight $\ell_2$-norm certified robustness guarantee for randomized smoothing with Gaussian noise. Other studies include Pinot et al. (2019); Carmon et al. (2019); Salman et al. (2019); Zhai et al. (2020); Dvijotham et al. (2019); Blum et al. (2020); Levine & Feizi (2020); Kumar et al. (2020); Yang et al. (2020); Zhang et al. (2020); Salman et al. (2020); Zheng et al. (2020). All these studies focused on top-1 predictions. Jia et al. (2020) derived the first $\ell_2$-norm certified robustness of top-$k$ predictions against adversarial perturbations for randomized smoothing with Gaussian noise and proved its tightness.

**Randomized smoothing for $\ell_0$-norm robustness:** All the above randomized smoothing based provable defenses were not (specifically) designed to certify $\ell_0$-norm robustness. They can be transformed to $\ell_0$-norm robustness via leveraging the relationship between $\ell_p$ norms. However, such transformations lead to suboptimal $\ell_0$-norm certified robustness. In response, multiple studies (Lee et al., 2019; Levine & Feizi, 2019; Dvijotham et al., 2019; Bojchevski et al., 2020; Jia et al., 2020; Zhang et al., 2021; Wang et al., 2021; Liu et al., 2021) proposed new randomized smoothing schemes to certify $\ell_0$-norm robustness. For instance, Lee et al. (2019) derived an $\ell_0$-norm certified robustness for classifiers with discrete features using randomized smoothing. In particular, for each feature, they keep its value with a certain probability and change it to a random value in the feature domain with an equal probability. Levine & Feizi (2019) proposed randomized ablation, which achieves state-of-the-art $\ell_0$-norm certified robustness. However, their work focused on top-1 predictions and they did not analyze the tightness of the certified robustness guarantee for top-1 predictions. We derive an almost tight $\ell_0$-norm certified robustness guarantee of top-$k$ predictions for randomized ablation.

## 5 CONCLUSION

In this work, we derive an almost tight $\ell_0$-norm certified robustness guarantee of top-$k$ predictions against adversarial perturbations for randomized ablation. We show that a label $l$ is provably among the top-$k$ labels predicted by a classifier smoothed by randomized ablation for a testing input when an attacker arbitrarily modifies a bounded number of features of the testing input. Moreover, we prove our derived bound is almost tight. Our empirical results show that our $\ell_0$-norm certified robustness is substantially better than those transformed from $\ell_2$-norm certified robustness. Interesting future works include exploring other noise to certify $\ell_0$-norm robustness for top-$k$ predictions and incorporating the information of the base classifier to derive larger certified radiuses.

### ACKNOWLEDGMENTS

We thank the anonymous reviewers for insightful reviews. This work was supported by the National Science Foundation under Grants No. 1937786 and 2125977, as well as the Army Research Office under Grant No. W911NF2110182.

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

## A   PROOF OF THEOREM 1

We define the following two random variables:

$$U = h(\mathbf{x}, e), V = h(\mathbf{x} + \delta, e). \tag{9}$$

where $U$ and $V$ denote the ablated inputs derived from $\mathbf{x}$ and its perturbed version $\mathbf{x} + \delta$ with parameter $e$, respectively. We use $\mathcal{S}$ to denote the domain space of $U$ and $V$.

Our proof is based on the Neyman-Pearson Lemma (Neyman & Pearson, 1933), and we present it as follows:

**Lemma 1** (Neyman-Pearson Lemma). *Suppose $U$ and $V$ are two random variables in the space $\mathcal{S}$ with probability distributions $\rho_u$ and $\rho_v$, respectively. Let $F : \mathcal{S} \to \{0, 1\}$ be a random or deterministic function. Then, we have the following:*

- *If $Z_1 = \{\mathbf{s} \in \mathcal{S} : \rho_u(\mathbf{s}) > \mu \cdot \rho_v(\mathbf{s})\}$ and $Z_2 = \{\mathbf{s} \in \mathcal{S} : \rho_u(\mathbf{s}) = \mu \cdot \rho_v(\mathbf{s})\}$ for some $\mu > 0$. Let $Z = Z_1 \cup Z_3$, where $Z_3 \subseteq Z_2$. If we have $Pr(F(U) = 1) \geq Pr(U \in Z)$, then $Pr(F(V) = 1) \geq Pr(V \in Z)$.*

- *If $Z_1 = \{\mathbf{s} \in \mathcal{S} : \rho_u(\mathbf{s}) < \mu \cdot \rho_v(\mathbf{s})\}$ and $Z_2 = \{\mathbf{s} \in \mathcal{S} : \rho_u(\mathbf{s}) = \mu \cdot \rho_v(\mathbf{s})\}$ for some $\mu > 0$. Let $Z = Z_1 \cup Z_3$, where $Z_3 \subseteq Z_2$. If we have $Pr(F(U) = 1) \leq Pr(U \in Z)$, then $Pr(F(V) = 1) \leq Pr(V \in Z)$.*

*Proof.* We show the proof of the first part, and the second part can be proved similarly. For simplicity, we use $F(1|\mathbf{s})$ and $F(0|\mathbf{s})$ to denote the conditional probabilities that $F(\mathbf{s}) = 0$ and $F(\mathbf{s}) = 1$, respectively. We use $Z^c$ to denote the complement of $Z$, i.e., $Z^c = \mathcal{S} \setminus Z$. We have the following:

$$\Pr(F(V) = 1) - \Pr(V \in Z) \tag{10}$$

$$= \sum_{\mathbf{s} \in \mathcal{S}} F(1|\mathbf{s}) \cdot \rho_v(\mathbf{s}) - \sum_{\mathbf{s} \in Z} \rho_v(\mathbf{s}) \tag{11}$$

$$= \sum_{\mathbf{s} \in Z^c} F(1|\mathbf{s}) \cdot \rho_v(\mathbf{s}) + \sum_{\mathbf{s} \in Z} F(1|\mathbf{s}) \cdot \rho_v(\mathbf{s}) - \sum_{\mathbf{s} \in Z} F(1|\mathbf{s}) \cdot \rho_v(\mathbf{s}) - \sum_{\mathbf{s} \in Z} F(0|\mathbf{s}) \cdot \rho_v(\mathbf{s}) \tag{12}$$

$$= \sum_{\mathbf{s} \in Z^c} F(1|\mathbf{s}) \cdot \rho_v(\mathbf{s}) - \sum_{\mathbf{s} \in Z} F(0|\mathbf{s}) \cdot \rho_v(\mathbf{s}) \tag{13}$$

$$\geq \frac{1}{\mu} \cdot \left( \sum_{\mathbf{s} \in Z^c} F(1|\mathbf{s}) \cdot \rho_u(\mathbf{s}) - \sum_{\mathbf{s} \in Z} F(0|\mathbf{s}) \cdot \rho_u(\mathbf{s}) \right) \tag{14}$$

$$= \frac{1}{\mu} \cdot \left( \sum_{\mathbf{s} \in Z^c} F(1|\mathbf{s}) \cdot \rho_u(\mathbf{s}) + \sum_{\mathbf{s} \in Z} F(1|\mathbf{s}) \cdot \rho_u(\mathbf{s}) - \sum_{\mathbf{s} \in Z} F(1|\mathbf{s}) \cdot \rho_u(\mathbf{s}) - \sum_{\mathbf{s} \in Z} F(0|\mathbf{s}) \cdot \rho_u(\mathbf{s}) \right) \tag{15}$$

$$= \frac{1}{\mu} \cdot \left( \sum_{\mathbf{s} \in \mathcal{S}} F(1|\mathbf{s}) \cdot \rho_u(\mathbf{s}) - \sum_{\mathbf{s} \in Z} \rho_u(\mathbf{s}) \right) \tag{16}$$

$$= \frac{1}{\mu} \cdot (\Pr(F(U) = 1) - \Pr(U \in Z)) \tag{17}$$

$$\geq 0. \tag{18}$$

We obtain (14) from (13) because $\rho_u(\mathbf{s}) \geq \mu \cdot \rho_v(\mathbf{s}), \forall \mathbf{s} \in Z$ and $\rho_u(\mathbf{s}) \leq \mu \cdot \rho_v(\mathbf{s}), \forall \mathbf{s} \in Z^c$. We have the last inequality because $\Pr(F(U) = 1) \geq \Pr(U \in Z)$. $\square$

Next, we will derive our certified robustness guarantee. For simplicity, we denote $\Gamma = \{1, 2, \cdots, c\} \setminus \{l\}$, i.e., $\Gamma$ denotes the set of all labels except $l$. We use $\Gamma_k$ to denote a set of $k$ labels in $\Gamma$.

**Calibrating the lower and upper bounds:** Recall that $p_l$ and $p_j, \forall j \neq l$ are integer multiple of $\frac{1}{\binom{d}{e}}$. Then, given the probability lower and upper bounds in Equation (3), we have the following:

$$\underline{p_l'} \triangleq \frac{\lceil \underline{p_l} \cdot \binom{d}{e} \rceil}{\binom{d}{e}} \leq \Pr(f(U) = l), \overline{p_j'} \triangleq \frac{\lfloor \overline{p_j} \cdot \binom{d}{e} \rfloor}{\binom{d}{e}} \geq \Pr(f(U) = j), \forall j \neq l, \tag{19}$$

**Deriving a lower bound of $\mathbf{Pr}(f(V) = l)$:** We will derive a lower bound of the probability $\Pr(f(V) = l)$. For simplicity, we define the following regions:

$$\mathcal{A} = \{\mathbf{s} \in \mathcal{S} | \mathbf{s} \preceq \mathbf{x}, \mathbf{s} \npreceq \mathbf{x} + \delta\}, \mathcal{B} = \{\mathbf{s} \in \mathcal{S} | \mathbf{s} \npreceq \mathbf{x}, \mathbf{s} \preceq \mathbf{x} + \delta\}, \mathcal{C} = \{\mathbf{s} \in \mathcal{S} | \mathbf{s} \preceq \mathbf{x}, \mathbf{s} \preceq \mathbf{x} + \delta\}, \tag{20}$$

where we say $\mathbf{s} \preceq \mathbf{x}$ if $\Pr(h(\mathbf{x}, e) = \mathbf{s}) > 0$, and we say $\mathbf{s} \npreceq \mathbf{x}$ if $\Pr(h(\mathbf{x}, e) = \mathbf{s}) = 0$. Intuitively, the notations $\preceq$ and $\npreceq$ mean that an ablated input can or cannot be derived from an input, respectively. For instance, region $\mathcal{A}$ contains ablated inputs that can be derived from $\mathbf{x}$ but cannot be derived from $\mathbf{x} + \delta$, region $\mathcal{B}$ contains ablated inputs that can be derived from $\mathbf{x} + \delta$ but cannot be derived from $\mathbf{x}$, and region $\mathcal{C}$ contains ablated inputs that can be derived from both $\mathbf{x}$ and $\mathbf{x} + \delta$. Suppose we have $r = ||\delta||_0$. Then, the size of $\mathcal{C}$ would be $\binom{d-r}{e}$ since $d - r$ features are the same for $\mathbf{x}$ and $\mathbf{x} + \delta$. Similarly, we know the size of $\mathcal{A}$ and $\mathcal{B}$ would be $\binom{d}{e} - \binom{d-r}{e}$. Since we keep $e$ features randomly sampled from $\mathbf{x}$ or $\mathbf{x} + \delta$ without replacement and set the remaining features to a special value, we have the following probability mass functions:

$$\Pr(U = \mathbf{s}) = \begin{cases} \frac{1}{\binom{d}{e}}, & \text{if } \mathbf{s} \in \mathcal{A} \cup \mathcal{C} \\ 0, & \text{otherwise.} \end{cases} \tag{21}$$

$$\Pr(V = \mathbf{s}) = \begin{cases} \frac{1}{\binom{d}{e}}, & \text{if } \mathbf{s} \in \mathcal{B} \cup \mathcal{C} \\ 0, & \text{otherwise.} \end{cases} \tag{22}$$

Since we know the size of $\mathcal{A}$, $\mathcal{B}$, and $\mathcal{C}$, as well as the probability mass functions of the random variables $U$ and $V$ in these regions, we have the following probabilities:

$$\Pr(U \in \mathcal{C}) = \frac{\binom{d-r}{e}}{\binom{d}{e}}, \Pr(U \in \mathcal{A}) = 1 - \frac{\binom{d-r}{e}}{\binom{d}{e}}, \Pr(U \in \mathcal{B}) = 0, \tag{23}$$

$$\Pr(V \in \mathcal{C}) = \frac{\binom{d-r}{e}}{\binom{d}{e}}, \Pr(V \in \mathcal{B}) = 1 - \frac{\binom{d-r}{e}}{\binom{d}{e}}, \Pr(V \in \mathcal{A}) = 0. \tag{24}$$

We consider the case of $\underline{p'_l} \geq 1 - \frac{\binom{d-r}{e}}{\binom{d}{e}}$. Note that we can do this because we aim to find a sufficient condition. We let $\mathcal{A}' \subseteq \mathcal{C}$ such that it satisfies the following:

$$\Pr(U \in \mathcal{A}') = \underline{p'_l} - \Pr(U \in \mathcal{A}). \tag{25}$$

Given region $\mathcal{A}'$, we construct the following region:

$$\mathcal{E} = \mathcal{A}' \cup \mathcal{A}. \tag{26}$$

Then, we have the following probability based on Equation (25):

$$\Pr(U \in \mathcal{E}) = \Pr(U \in \mathcal{A}) + \Pr(U \in \mathcal{A}') = \underline{p'_l}. \tag{27}$$

We define a binary function $F(\mathbf{s}) = \mathbb{I}(f(\mathbf{s}) = l)$. Then, we have the following:

$$\Pr(F(U) = 1) = \Pr(f(U) = l) \geq \underline{p'_l} = \Pr(U \in \mathcal{E}). \tag{28}$$

The middle inequality is based on Equation (19) and the right-hand equality is from Equation (27). Furthermore, we have $\Pr(U = \mathbf{s}) > 1 \cdot \Pr(V = \mathbf{s})$ if and only if $\mathbf{s} \in \mathcal{A}$, and $\Pr(U = \mathbf{s}) = 1 \cdot \Pr(V = \mathbf{s})$ if $\mathbf{s} \in \mathcal{A}'$. Therefore, we can apply Lemma 1 and we have the following:

$$\Pr(F(V) = 1) = \Pr(f(V) = l) \geq \Pr(V \in \mathcal{E}). \tag{29}$$

Therefore, we have the following lower bound for $\Pr(f(V) = l)$:

$$\Pr(V \in \mathcal{E}) \tag{30}$$

$$= \Pr(V \in \mathcal{A}') + \Pr(V \in \mathcal{A}) \tag{31}$$

$$= \Pr(V \in \mathcal{A}') \tag{32}$$

$$= \Pr(U \in \mathcal{A}') \tag{33}$$

$$=\underline{p}'_l - (1 - \frac{\binom{d-r}{e}}{\binom{d}{e}}). \tag{34}$$

Note that we have Equation (34) from (33) based on Equation (25).

**Deriving an upper bound of** $\max_{\Gamma_k \subset \Gamma} \min_{j \in \Gamma_k} \mathbf{Pr}(f(V) = j)$**:** We use $\Lambda$ to denote an arbitrary subset of $\Gamma_k$, i.e., $\Lambda \subseteq \Gamma_k$. We denote $\overline{p}'_\Lambda = \sum_{j \in \Lambda} \overline{p}'_j$, which is the sum of the upper bound of the probability for the labels in $\Lambda$. We assume $\overline{p}'_\Lambda \leq \Pr(U \in \mathcal{C})$. We can make this assumption because we aim to find a sufficient condition. Given $\overline{p}'_\Lambda$, we can find a region $\mathcal{H}_\Lambda \subseteq \mathcal{C}$ such that we have the following:

$$\overline{p}'_\Lambda = \Pr(U \in \mathcal{H}_\Lambda). \tag{35}$$

Given the region $\mathcal{H}_\Lambda$, we construct the following region:

$$\mathcal{I}_\Lambda = \mathcal{H}_\Lambda \cup \mathcal{B}. \tag{36}$$

Then, we have the following probability:

$$\Pr(U \in \mathcal{I}_\Lambda) = \Pr(U \in \mathcal{H}_\Lambda) + \Pr(U \in \mathcal{B}) = \overline{p}'_\Lambda. \tag{37}$$

Furthermore, for any given $\Lambda$, we define a binary function $G(\mathbf{s}) = \mathbb{I}(f(\mathbf{s}) \in \Lambda)$. Then, we have the following:

$$\Pr(G(U) = 1) = \Pr(f(U) \in \Lambda) = \sum_{j \in \Lambda} \Pr(f(U) = j) \leq \overline{p}'_\Lambda = \Pr(U \in \mathcal{I}_\Lambda). \tag{38}$$

We have $\sum_{j \in \Lambda} \Pr(f(U) = j) \leq \overline{p}'_\Lambda$ based on Equation (19) and we have rightmost equality from Equation (37). Then, we can apply Lemma 1 and we have the following:

$$\Pr(G(V) = 1) \leq \Pr(V \in \mathcal{I}_\Lambda). \tag{39}$$

The value of $\Pr(V \in \mathcal{I}_\Lambda)$ can be computed as follows:

$$\Pr(V \in I_\Lambda) \tag{40}$$
$$=\Pr(V \in \mathcal{H}_\Lambda) + \Pr(V \in \mathcal{B}) \tag{41}$$
$$=\Pr(U \in \mathcal{H}_\Lambda) + (1 - \frac{\binom{d-r}{e}}{\binom{d}{e}}) \tag{42}$$
$$=\overline{p}'_\Lambda + (1 - \frac{\binom{d-r}{e}}{\binom{d}{e}}), \tag{43}$$

where the last equality is from Equation (35). Therefore, we have the following:

$$\sum_{j \in \Lambda} \Pr(f(V) = j) \tag{44}$$
$$=\Pr(f(V) \in \Lambda) \tag{45}$$
$$=\Pr(G(V) = 1) \tag{46}$$
$$\leq\Pr(V \in \mathcal{I}_\Lambda) \tag{47}$$
$$=\overline{p}_\Lambda + (1 - \frac{\binom{d-r}{e}}{\binom{d}{e}}). \tag{48}$$

Moreover, we have the following:

$$\min_{j \in \Gamma_k} \Pr(f(V) = j) \leq \min_{j \in \Lambda} \Pr(f(V) = j) \leq \frac{\sum_{j \in \Lambda} \Pr(f(V) = j)}{|\Lambda|} = \frac{\Pr(f(V) \in \Lambda)}{|\Lambda|}. \tag{49}$$

We have the leftmost inequality because $\Lambda \subseteq \Gamma_k$, and we have the middle inequality because the smallest value in a set is no larger than the average value of the set. Taking all possible $\Lambda$ into consideration and we have the following:

$$\min_{j \in \Gamma_k} \Pr(f(V) = j) \tag{50}$$

$$\leq \min_{\Lambda \subseteq \Gamma_k} \frac{\Pr(f(V) \in \Lambda)}{|\Lambda|} \tag{51}$$

$$= \min_{t=1}^{k} \min_{\Lambda \subseteq \Gamma_k, |\Lambda| = t} \frac{\Pr(f(V) \in \Lambda)}{|\Lambda|} \tag{52}$$

$$= \min_{t=1}^{k} \frac{\Pr(f(V) \in \Upsilon_t)}{t} \tag{53}$$

$$\leq \min_{t=1}^{k} \frac{\overline{p}'_{\Upsilon_t} + (1 - \frac{\binom{d-r}{e}}{\binom{d}{e}})}{t}, \tag{54}$$

where $\Upsilon_t$ is the set of $t$ labels in $\Gamma_k$ whose probability upper bounds are the smallest, where ties are broken uniformly at random. The upper bound of $\Pr(f(V) \in \Upsilon_t)$ is increasing as $\overline{p}'_{\Upsilon_t}$ increases. Therefore, the upper bound of $\frac{\Pr(f(V) \in \Upsilon_t)}{t}$ reaches the maximum value when $\Gamma_k = \{a_1, a_2, \cdots, a_k\}$, i.e., $\Gamma_k$ is the set of labels in $\Gamma$ with the largest probability upper bounds. In other words, we have the following:

$$\max_{\Gamma_k \subset \Gamma} \min_{j \in \Gamma_k} \Pr(f(V) = j) \leq \min_{t=1}^{k} \frac{\overline{p}'_{\Upsilon_t} + (1 - \frac{\binom{d-r}{e}}{\binom{d}{e}})}{t}, \tag{55}$$

where $\Upsilon_t = \{a_1, a_2, \cdots, a_t\}$.

**Deriving the certified radius:** Our goal is to make $\Pr(f(V) = l) > \max_{\Gamma_k \subset \Gamma} \min_{j \in \Gamma_k} \Pr(f(V) = j)$. Therefore, it is sufficient to satisfy the following:

$$\underline{p}'_l - (1 - \frac{\binom{d-r}{e}}{\binom{d}{e}}) > \min_{t=1}^{k} \frac{\overline{p}'_{\Upsilon_t} + (1 - \frac{\binom{d-r}{e}}{\binom{d}{e}})}{t}, \tag{56}$$

where $\Upsilon_t = \{a_1, a_2, \cdots, a_t\}$. Therefore, we can find the maximum $r$ that satisfies the above condition. Formally, we can solve the following optimization problem to find $r_l$:

$$r_l = \arg\max_r r \tag{57}$$

$$\text{s.t. } \underline{p}'_l - (1 - \frac{\binom{d-r}{e}}{\binom{d}{e}}) > \min_t \frac{\overline{p}'_{\Upsilon_t} + (1 - \frac{\binom{d-r}{e}}{\binom{d}{e}})}{t}, \tag{58}$$

where $\Upsilon_t = \{a_1, a_2, \cdots, a_t\}$. Note that we make two assumptions in our derivation, i.e., $\underline{p}'_l \geq (1 - \frac{\binom{d-r}{e}}{\binom{d}{e}})$ and $\overline{p}'_{\Upsilon_t} \leq \frac{\binom{d-r}{e}}{\binom{d}{e}}$. In particular, when Equation (58) is satisfied, we must have $\underline{p}'_l \geq (1 - \frac{\binom{d-r}{e}}{\binom{d}{e}})$ since the left-hand side of Equation (58) is non-negative. In addition, we have $\underline{p}'_l + \overline{p}'_{\Upsilon_t} \leq 1$ in practice. Therefore, we have $\overline{p}'_{\Upsilon_t} \leq 1 - \underline{p}'_l \leq \frac{\binom{d-r}{e}}{\binom{d}{e}}$.

**Technical differences with Jia et al. (2020):** Our technical contribution in proving the theorem is the construction of new discrete regions such that the Neyman-Pearson Lemma can be used. Our proof of Theorem 1 has the following differences with Jia et al. First, the construct of the regions $\mathcal{A}/\mathcal{B}/\mathcal{C}$ (Eq 18) are different from Jia et al. due to the discrete space. Second, deriving the lower bound of $\mathbf{Pr}(f(V) = l)$ faces two new challenges in our case. The first challenge is that we need to find two regions $\mathcal{A}$ and $\mathcal{A}'$ while Jia et al. just need to find one region. The second challenge is how to find these regions. To address the challenge, we first take into consideration of whether $\mathcal{A}'$ exists or not. If $\mathcal{A}'$ exists, we need to shrink region $\mathcal{A}'$ because our space is discrete. These two challenges do not exist in the continuous case considered by Jia et al. Third, similar to deriving the lower bound of $\mathbf{Pr}(f(V) = l)$, our work is also different from Jia et al. at deriving the upper bound of $\max_{\Gamma_k \in \Gamma} \min_{j \in \Gamma_k} \mathbf{Pr}(f(V) = j)$.

## B    PROOF OF THEOREM 2

We consider two scenarios: $k = 1$ and $k \neq 1$. In particular, we first consider the scenario where $k = 1$. We have $\Gamma_1 = \{a_1\}$ when $k = 1$. We consider two cases.

**Case I:** In this case, we consider $\underline{p'_l} < (1 - \frac{\binom{d-r_l-1}{e}}{\binom{d}{e}})$. We let $\mathcal{A}_l \subseteq \mathcal{A}$ be the region that satisfies the following:

$$\underline{p'_l} = \Pr(U \in \mathcal{A}_l). \tag{59}$$

We can find such region because $\underline{p'_l}$ is an integer multiple of $\frac{1}{\binom{d}{e}}$. We let $\mathcal{D}_l = \mathcal{A}_l$ and we have the following:

$$\underline{p'_l} = \Pr(U \in \mathcal{D}_l), \Pr(V \in \mathcal{D}_l) = 0. \tag{60}$$

Then, we can divide the remaining region $(\mathcal{A} \cup \mathcal{C}) \setminus \mathcal{D}_l$ into $c - 1$ disjoint regions such that we have the following:

$$\forall j \in \{1, 2, \cdots, c\} \setminus \{l\}, \Pr(U \in \mathcal{D}_j) \leq \overline{p'_j}. \tag{61}$$

We can find these regions because we have $\underline{p'_l} + \sum_{s \neq l} \overline{p'_s} \geq 1$. Moreover, we have the following:

$$\forall j \in \{1, 2, \cdots, c\} \setminus \{l\}, \Pr(V \in \mathcal{D}_j) \geq 0. \tag{62}$$

Given these regions, we construct the following base classifier:

$$f^*(\mathbf{z}) = j, \text{ if } \mathbf{z} \in \mathcal{D}_j. \tag{63}$$

Note that $f^*$ is well defined and is consistent with Equation (3). It is easy to see that label $l$ is not the predicted label by the corresponding smoothed classifier $g^*$ when $\|\delta\|_0 > r_l$.

**Case II:** In this case, we consider $\underline{p'_l} \geq (1 - \frac{\binom{d-r_l-1}{e}}{\binom{d}{e}})$. Since $r_l$ is the maximum value that satisfies Equation (6), we have the following condition when $\|\delta\|_0 = r_l + 1$:

$$\underline{p'_l} - (1 - \frac{\binom{d-r_l-1}{e}}{\binom{d}{e}}) \leq \overline{p'_{a_1}} + (1 - \frac{\binom{d-r_l-1}{e}}{\binom{d}{e}}). \tag{64}$$

We let $\mathcal{A}_l = \mathcal{A}$ and we can find $\mathcal{C}_l \in \mathcal{C}$ such that the following equation holds:

$$\Pr(U \in \mathcal{C}_l) = \underline{p'_l} - (1 - \frac{\binom{d-r_l-1}{e}}{\binom{d}{e}}). \tag{65}$$

Then, we let $\mathcal{D}_l = \mathcal{C}_l \cup \mathcal{A}_l$ and we have the following:

$$\Pr(U \in \mathcal{D}_l) = \underline{p'_l}. \tag{66}$$

Furthermore, we have the following:

$$\Pr(V \in \mathcal{D}_l) \tag{67}$$
$$= \Pr(V \in \mathcal{C}_l) + \Pr(V \in \mathcal{A}_l) \tag{68}$$
$$= \Pr(U \in \mathcal{C}_l) + 0 \tag{69}$$
$$= \underline{p'_l} - (1 - \frac{\binom{d-r_l-1}{e}}{\binom{d}{e}}), \tag{70}$$

where the last equality is from Equation (65). Since we have $\underline{p'_l} + \overline{p'_{a_1}} \leq 1$, we can find region $\mathcal{C}_{a_1} \in \mathcal{C} \setminus \mathcal{C}_l$ such that we have the following:

$$\Pr(U \in \mathcal{C}_{a_1}) = \overline{p'_{a_1}}. \tag{71}$$

We define $\mathcal{D}_{a_1} = \mathcal{C}_{a_1} \cup \mathcal{B}$. Then, we have $\Pr(U \in \mathcal{D}_{a_1}) = \Pr(U \in \mathcal{C}_{a_1}) = \overline{p'_{a_1}}$. Similarly, we have the following:

$$\Pr(V \in \mathcal{D}_{a_1}) \tag{72}$$
$$= \Pr(V \in \mathcal{C}_{a_1}) + \Pr(V \in \mathcal{B}) \tag{73}$$
$$= \overline{p'_{a_1}} + (1 - \frac{\binom{d-r_l-1}{e}}{\binom{d}{e}}). \tag{74}$$

Finally, we can divide the remaining region $\mathcal{A} \cup \mathcal{C} \setminus (\mathcal{D}_l \cup \mathcal{C}_{a_1})$ into $c - 2$ disjoint regions such that we have the following:

$$\forall j \in \{1, 2, \cdots, c\} \setminus (\{l\} \cup \{a_1\}), \Pr(U \in \mathcal{D}_j) \leq \overline{p}'_j. \tag{75}$$

We can find these region because $\underline{p}'_l + \sum_{s \neq l} \overline{p}'_s \geq 1$. Given these regions, we construct the following base classifier:

$$f^*(\mathbf{z}) = j, \text{ if } \mathbf{z} \in \mathcal{D}_j. \tag{76}$$

Note that $f^*$ is well defined and is consistent with Equation (3). Next, we show that label $l$ is not in the top-1 predicted labels by the smoothed classifier or there exist ties when the $\ell_0$ perturbation is larger than $r_l$. In particular, we have the following:

$$\Pr(f^*(V) = a_1 | \, \|\delta\|_0 > r_l) \tag{77}$$

$$= \Pr(V \in \mathcal{D}_{a_1} | \, \|\delta\|_0 > r_l) \tag{78}$$

$$= \overline{p}'_{a_1} + (1 - \frac{\binom{d-r_l-1}{e}}{\binom{d}{e}}) \tag{79}$$

$$\geq \underline{p}'_l - (1 - \frac{\binom{d-r-1}{e}}{\binom{d}{e}}) \tag{80}$$

$$= \Pr(V \in \mathcal{D}_l | \, \|\delta\|_0 > r_l) \tag{81}$$

$$= \Pr(f^*(V) = l | \, \|\delta\|_0 > r_l). \tag{82}$$

We have Equation (80) from (79) based on Equation (64). Therefore, the label $l$ is not predicted by the corresponding smoothed classifier $g^*$ or there exist ties. Combining the two cases, we reach the conclusion.

Next, we will show our bound is almost tight when $k \neq 1$. In particular, we will show we can construct a classifier $f^*$ such that the label $l$ is not among the top-$k$ predicted labels or there exist ties when the adversarial perturbation is larger than $r_l + 1$. Similarly, we consider two cases.

**Case I:** In this case, we consider $\underline{p}'_l < (1 - \frac{\binom{d-r_l-2}{e}}{\binom{d}{e}})$. We let $\mathcal{A}_l \subseteq \mathcal{A}$ be the region that satisfies the following:

$$\underline{p}'_l = \Pr(U \in \mathcal{A}_l). \tag{83}$$

We can find such region because $\underline{p}'_l$ is an integer multiply of $\nu = \frac{1}{\binom{d}{e}}$. We let $\mathcal{D}_l = \mathcal{A}_l$ and we have the following:

$$\underline{p}'_l = \Pr(U \in \mathcal{D}_l), \Pr(V \in \mathcal{D}_l) = 0. \tag{84}$$

Then, we can divide the remaining region $(\mathcal{A} \cup \mathcal{C}) \setminus \mathcal{D}_l$ into $c - 1$ disjoint regions such that we have the following:

$$\forall j \in \{1, 2, \cdots, c\} \setminus \{l\}, \Pr(U \in \mathcal{D}_j) \leq \overline{p}'_j. \tag{85}$$

We can find these regions because we have $\underline{p}'_l + \sum_{s \neq l} \overline{p}'_s \geq 1$. Moreover, we have the following:

$$\forall j \in \{1, 2, \cdots, c\} \setminus \{l\}, \Pr(V \in \mathcal{D}_j) \geq 0. \tag{86}$$

Given these regions, we construct the following base classifier:

$$f^*(\mathbf{z}) = j, \text{ if } \mathbf{z} \in \mathcal{D}_j. \tag{87}$$

Note that $f^*$ is well defined and is consistent with Equation (3). It is easy to see that label $l$ is not among the top-$k$ predicted labels or there exist ties when $\|\delta\|_0 > r_l + 1$.

**Case II:** In this case, we consider $\underline{p}'_l \geq (1 - \frac{\binom{d-r_l-2}{e}}{\binom{d}{e}})$. For simplicity, we denote the following quantity:

$$\nu = \frac{1}{\binom{d}{e}}. \tag{88}$$

Since $r_l$ is the maximum value that satisfies Equation (6), we have the following condition:

$$\underline{p_l'} - (1 - \frac{\binom{d-r_l-1}{e}}{\binom{d}{e}}) \le \min_t \frac{\overline{p}_{\Upsilon_t}' + (1 - \frac{\binom{d-r_l-1}{e}}{\binom{d}{e}})}{t}. \tag{89}$$

In other words, the left-hand side of Equation (6) is no larger than its right-hand side when $r = r_l + 1$. Based on the recurrence relation of the binomial coefficient, we have the following:

$$\binom{d - r_l - 1}{e} = \binom{d - r_l - 2}{e} + \binom{d - r_l - 2}{e - 1}. \tag{90}$$

Combining with the condition $\binom{d-r_l-2}{e-1} \ge 1$, we have the following:

$$\underline{p_l'} - (1 - \frac{\binom{d-r_l-1}{e}}{\binom{d}{e}}) \tag{91}$$

$$= \underline{p_l'} - (1 - \frac{\binom{d-r_l-2}{e}}{\binom{d}{e}} - \frac{\binom{d-r_l-2}{e-1}}{\binom{d}{e}}) \tag{92}$$

$$= \underline{p_l'} + \frac{\binom{d-r_l-2}{e-1}}{\binom{d}{e}} - (1 - \frac{\binom{d-r_l-2}{e}}{\binom{d}{e}}) \tag{93}$$

$$\ge \underline{p_l'} + \nu - (1 - \frac{\binom{d-r_l-2}{e}}{\binom{d}{e}}). \tag{94}$$

Similarly, we have the following:

$$\min_t \frac{\overline{p}_{\Upsilon_t}' + (1 - \frac{\binom{d-r_l-1}{e}}{\binom{d}{e}})}{t} \tag{95}$$

$$= \min_t \frac{\overline{p}_{\Upsilon_t}' - \frac{\binom{d-r_l-2}{e-1}}{\binom{d}{e}} + (1 - \frac{\binom{d-r_l-2}{e}}{\binom{d}{e}})}{t} \tag{96}$$

$$\le \min_t \frac{\overline{p}_{\Upsilon_t}' - \nu + (1 - \frac{\binom{d-r_l-2}{e}}{\binom{d}{e}})}{t}. \tag{97}$$

Then, based on Equation (89), we have the following:

$$\underline{p_l'} + \nu - (1 - \frac{\binom{d-r_l-2}{e}}{\binom{d}{e}}) \le \min_t \frac{\overline{p}_{\Upsilon_t}' - \nu + (1 - \frac{\binom{d-r_l-2}{e}}{\binom{d}{e}})}{t} \tag{98}$$

$$\Longleftrightarrow \underline{p_l'} - (1 - \frac{\binom{d-r_l-2}{e}}{\binom{d}{e}}) \le \min_t \frac{\overline{p}_{\Upsilon_t}' - \nu - t \cdot \nu + (1 - \frac{\binom{d-r_l-2}{e}}{\binom{d}{e}})}{t} \tag{99}$$

$$\Longrightarrow \underline{p_l'} - (1 - \frac{\binom{d-r_l-2}{e}}{\binom{d}{e}}) < \min_t \frac{\overline{p}_{\Upsilon_t}' - t \cdot \nu + (1 - \frac{\binom{d-r_l-2}{e}}{\binom{d}{e}})}{t}. \tag{100}$$

For simplicity, we denote the following:

$$w = \underset{t=1}{\overset{k}{\arg\min}} \frac{\overline{p}_{\Upsilon_t}' - t \cdot \nu + (1 - \frac{\binom{d-r_l-2}{e}}{\binom{d}{e}})}{t}, \tag{101}$$

where ties are broken uniformly at random. Then, based on Equation (100), we have the following:

$$\underline{p_l'} - (1 - \frac{\binom{d-r_l-2}{e}}{\binom{d}{e}}) < \frac{\overline{p}_{\Upsilon_w}' - w \cdot \nu + (1 - \frac{\binom{d-r_l-2}{e}}{\binom{d}{e}})}{w}. \tag{102}$$

Given Equation (101), we have the following if $w < k$:

$$\frac{\overline{p}'_{\Upsilon_{w+1}} - (w+1)\cdot\nu + (1 - \frac{\binom{d-r_l-2}{e}}{\binom{d}{e}})}{w+1} \geq \frac{\overline{p}'_{\Upsilon_w} - w\cdot\nu + (1 - \frac{\binom{d-r_l-2}{e}}{\binom{d}{e}})}{w} \tag{103}$$

$$\Longleftrightarrow \frac{\overline{p}'_{\Upsilon_{w+1}} + (1 - \frac{\binom{d-r_l-2}{e}}{\binom{d}{e}})}{w+1} \geq \frac{\overline{p}'_{\Upsilon_w} + (1 - \frac{\binom{d-r_l-2}{e}}{\binom{d}{e}})}{w} \tag{104}$$

$$\Longleftrightarrow \overline{p}'_{\Upsilon_{w+1}} + (1 - \frac{\binom{d-r_l-2}{e}}{\binom{d}{e}}) \geq (w+1)\cdot\frac{\overline{p}'_{\Upsilon_w} + (1 - \frac{\binom{d-r_l-2}{e}}{\binom{d}{e}})}{w} \tag{105}$$

$$\Longleftrightarrow \overline{p}'_{\Upsilon_{w+1}} - \overline{p}'_{\Upsilon_w} \geq \frac{\overline{p}'_{\Upsilon_w} + (1 - \frac{\binom{d-r_l-2}{e}}{\binom{d}{e}})}{w} \tag{106}$$

$$\Longleftrightarrow \overline{p}'_{a_{w+1}} \geq \frac{\overline{p}'_{\Upsilon_w} + (1 - \frac{\binom{d-r_l-2}{e}}{\binom{d}{e}})}{w}, \tag{107}$$

where $\Upsilon_w = \{a_1, a_2, \cdots, a_w\}$. Similarly, we have the following if $w > 1$:

$$\frac{\overline{p}'_{\Upsilon_{w-1}} - (w-1)\cdot\nu + (1 - \frac{\binom{d-r_l-2}{e}}{\binom{d}{e}})}{w-1} \geq \frac{\overline{p}'_{\Upsilon_w} - w\nu + (1 - \frac{\binom{d-r_l-2}{e}}{\binom{d}{e}})}{w} \tag{108}$$

$$\Longleftrightarrow \frac{\overline{p}'_{\Upsilon_{w-1}} + (1 - \frac{\binom{d-r_l-2}{e}}{\binom{d}{e}})}{w-1} \geq \frac{\overline{p}'_{\Upsilon_w} + (1 - \frac{\binom{d-r_l-2}{e}}{\binom{d}{e}})}{w} \tag{109}$$

$$\Longleftrightarrow \overline{p}'_{\Upsilon_{w-1}} + (1 - \frac{\binom{d-r_l-2}{e}}{\binom{d}{e}}) \geq (w-1)\cdot\frac{\overline{p}'_{\Upsilon_w} + (1 - \frac{\binom{d-r_l-2}{e}}{\binom{d}{e}})}{w} \tag{110}$$

$$\Longleftrightarrow \overline{p}'_{a_w} \leq \frac{\overline{p}'_{\Upsilon_w} + (1 - \frac{\binom{d-r_l-2}{e}}{\binom{d}{e}})}{w}. \tag{111}$$

Note that the Equation (111) also holds when $w = 1$. Next, we will show we can build a base classifier $f^*$ such that the label $l$ is not in the top-$k$ predicted labels or there exist ties when the adversarial perturbation is larger than $r_l + 1$. Our proof relies on constructing disjoint regions for label $l$, $\Upsilon_k$, and $\{1, 2, \cdots, c\} \setminus (\{l\} \cup \Upsilon_k)$, respectively.

We let $\mathcal{A}_l = \mathcal{A}$ and we can find $\mathcal{C}_l \in \mathcal{C}$ such that the following equation holds:

$$\Pr(U \in \mathcal{C}_l) = \underline{p}'_l - (1 - \frac{\binom{d-r_l-2}{e}}{\binom{d}{e}}). \tag{112}$$

Then, we let $\mathcal{D}_l = \mathcal{C}_l \cup \mathcal{A}_l$ and we have the following:

$$\Pr(U \in \mathcal{D}_l) = \underline{p}'_l. \tag{113}$$

Furthermore, we have the following:

$$\Pr(V \in \mathcal{D}_l) \tag{114}$$
$$= \Pr(V \in \mathcal{C}_l) + \Pr(V \in \mathcal{A}_l) \tag{115}$$
$$= \Pr(U \in \mathcal{C}_l) + 0 \tag{116}$$
$$= \underline{p}'_l - (1 - \frac{\binom{d-r_l-2}{e}}{\binom{d}{e}}), \tag{117}$$

where the last equality is from Equation (112). For simplicity, we denote the following value:

$$\tau = \frac{\overline{p}'_{\Upsilon_w} + (1 - \frac{\binom{d-r_l-2}{e}}{\binom{d}{e}})}{w}. \tag{118}$$

Next, we will construct the region for $\forall j \in \Upsilon_w$. Based on Equation (112), we have the following:

$$\Pr(U \in \mathcal{C} \setminus \mathcal{C}_l) \tag{119}$$

$$=\Pr(U \in \mathcal{C}) - \Pr(U \in \mathcal{C}_l) \tag{120}$$

$$=(1 - \frac{\binom{d-r_l-2}{e}}{\binom{d}{e}}) - (\underline{p'_l} - (1 - \frac{\binom{d-r_l-2}{e}}{\binom{d}{e}})) \tag{121}$$

$$=1 - \underline{p'_l}. \tag{122}$$

For $\forall j \in \Upsilon_w$, we can find disjoint region $\mathcal{C}_j \subseteq \mathcal{C} \setminus \mathcal{C}_l$ such that we have the following:

$$\Pr(U \in \mathcal{C}_j) = \overline{p}'_j. \tag{123}$$

We can find these regions because the summation of the probability of $U$ in these regions is less than the probability of $U$ in $\mathcal{C} \setminus \mathcal{C}_l$, i.e., we have the following:

$$\sum_{j \in \Upsilon_w} \overline{p}'_j = \overline{p}'_{\Upsilon_w} \leq 1 - \underline{p'_l} = \Pr(U \in \mathcal{C} \setminus \mathcal{C}_l), \tag{124}$$

where the middle inequality is from the condition $\underline{p'_l} + \sum_{s \in \Upsilon_k} \overline{p}'_s \leq 1$, and the right equality is based on Equation (119) - (122). Given these regions, we have the following:

$$\forall j \in \Upsilon_w, \Pr(V \in \mathcal{C}_j) = \overline{p}'_j. \tag{125}$$

Based on Equation (111), definition of $\tau$ in Equation (118), and $\forall j \in \Upsilon_w, \overline{p}'_j \leq \overline{p}'_{a_w}$, we have the following:

$$\forall j \in \Upsilon_w, \overline{p}'_j \leq \overline{p}'_{a_w} \leq \tau. \tag{126}$$

Then, for $\forall j \in \Upsilon_w$, we can find disjoint region $\mathcal{B}_j \in \mathcal{B}$ such that we have the following:

$$\tau - \nu - \overline{p}'_j \leq \Pr(V \in \mathcal{B}_j) \leq \tau - \overline{p}'_j. \tag{127}$$

We can construct these regions for three reasons: 1) the value of $\tau - \overline{p}'_j$ is no smaller than 0 based on Equation (126), 2) $\forall j \in \Upsilon_w$, there exists a number in the range $[\tau - \nu - \overline{p}'_j, \tau - \overline{p}'_j]$ that is an integer multiple of $\frac{1}{\binom{d}{e}}$, and 3) the summation of the probability of $V$ in these regions is no larger than the probability of $V$ in $\mathcal{B}$, i.e., we have the following:

$$\sum_{j \in \Upsilon_w} \Pr(V \in \mathcal{B}_j) \tag{128}$$

$$\leq \sum_{j \in \Upsilon_w} (\tau - \overline{p}'_j) \tag{129}$$

$$=\overline{p}'_{\Upsilon_w} + (1 - \frac{\binom{d-r_l-2}{e}}{\binom{d}{e}}) - \overline{p}'_{\Upsilon_w} \tag{130}$$

$$\leq (1 - \frac{\binom{d-r_l-2}{e}}{\binom{d}{e}}) \tag{131}$$

$$=\Pr(V \in \mathcal{B}). \tag{132}$$

For $\forall j \in \Upsilon_w$, we let $\mathcal{D}_j = \mathcal{C}_j \cup \mathcal{B}_j$. Then, we have the following:

$$\Pr(V \in \mathcal{D}_j) \tag{133}$$

$$=\Pr(V \in \mathcal{C}_j) + \Pr(V \in \mathcal{B}_j) \tag{134}$$

$$\geq \overline{p}'_j + \tau - \nu - \overline{p}'_j \tag{135}$$

$$=\tau - \nu, \tag{136}$$

where the Equation (135) from (134) is based on Equation (123) and (127). Next, we will construct the regions for the labels in $\Upsilon_k \setminus \Upsilon_w$. In particular, for $\forall j \in \{a_{w+1}, a_{w+2}, \cdots, a_k\}$, we can find disjoint region $\mathcal{D}_j \in \mathcal{C} \setminus (\mathcal{C}_l \cup (\cup_{s \in \Upsilon_w} \mathcal{C}_s))$ such that we have the following:

$$\Pr(U \in \mathcal{D}_j) = \overline{p}'_j. \tag{137}$$

Note that we can find these regions because $\underline{p}'_l + \sum_{s \in \Upsilon_k} \overline{p}'_s \leq 1$. Similarly, we have the following for $\forall j \in \Upsilon_k \setminus \Upsilon_w$:

$$\Pr(V \in \mathcal{D}_j) = \overline{p}'_j \geq \tau. \tag{138}$$

We have the left inequality because $\forall j \in \Upsilon_k \setminus \Upsilon_w, \overline{p}'_j \geq \tau$ based on Equation (107). Finally, we can divide the remaining region $\mathcal{D}_j \subseteq \mathcal{C} \cup \mathcal{A} \setminus (\mathcal{D}_l \cup (\cup_{s \in \Upsilon_k} \mathcal{C}_s))$ into $c - k - 1$ disjoint regions such that we have the following:

$$\forall j \in \{1, 2, \cdots, c\} \setminus (\{l\} \cup \Upsilon_k), \Pr(U \in \mathcal{D}_j) \leq \overline{p}'_j. \tag{139}$$

We can find these region because $\underline{p}'_l + \sum_{s \neq l} \overline{p}'_s \geq 1$. Given these regions, we construct the following base classifier:

$$f^*(\mathbf{z}) = j, \text{ if } \mathbf{z} \in \mathcal{D}_j. \tag{140}$$

Note that $f^*$ is well defined and is consistent with Equation (3). Next, we show that label $l$ is not in the top-$k$ predicted labels by the smoothed classifier when the $\ell_0$ perturbation is larger than $r_l + 1$. In particular, for $\forall j \in \Upsilon_k$, we have the following:

$$\Pr(f^*(V) = j | \|\delta\|_0 > r_l + 1) \tag{141}$$
$$=\Pr(V \in \mathcal{D}_j | \|\delta\|_0 > r_l + 1) \tag{142}$$
$$\geq \tau - \nu \tag{143}$$
$$=\frac{\overline{p}'_{\Upsilon_w} - w \cdot \nu + (1 - \frac{\binom{d-r_l-2}{e}}{\binom{d}{e}})}{w} \tag{144}$$
$$>\underline{p}'_l - (1 - \frac{\binom{d-r-2}{e}}{\binom{d}{e}}) \tag{145}$$
$$\geq \Pr(V \in \mathcal{D}_l | \|\delta\|_0 > r_l + 1) \tag{146}$$
$$=\Pr(f^*(V) = l | \|\delta\|_0 > r_l + 1). \tag{147}$$

We have Equation (145) from (144) based on Equation (102). Therefore, the label $l$ is not among the top-$k$ predicted labels by the corresponding smoothed classifier $g^*$. Combining the two cases, we reach the conclusion.

**Takeaway:** The key challenge in proving the tightness of certified robustness guarantee is that we are not able to find regions in the discrete space that satisfy certain conditions. The reason is that we cannot arbitrarily divide the discrete space into different regions. To address the challenge, we propose to relax the conditions in finding the regions to prove that the certified robustness guarantee is almost tight. The idea in our proof is very general and we hope our proof can inspire future research in proving the (almost) tightness for $\ell_0$-norm certified robustness guarantee.

## C   OTHER APPLICATIONS

In this paper, we focus on image classification. However, our method is also applicable for other applications such as graph neural networks and 3D deep learning. For example, we conduct experiments for 3D deep learning. In particular, we evaluate our methods on the ModelNet40 (Wu et al., 2015) benchmark dataset and use PointNet (Qi et al., 2017) as the base classifier. We set $e = 16$, $n = 10,000$, and $\alpha = 0.001$. When the number of modified points is 10, 20, 30, 40, and 50, the certified accuracies for top-1 prediction are 0.779, 0.743, 0.700, 0.649, and 0.570; and the certified accuracies for top-3 prediction are 0.901, 0.867, 0.829, 0.803, and 0.775.

## D   COMPARING WITH CHIANG ET AL. (2020)

We also compare with our method with Chiang et al. (2020). We note that the method in Chiang et al. is only applicable for top-1 prediction. We compare with the method on the CIFAR10 dataset for top-1 predictions using the publicly available code. The results are as follows. When the number of perturbed pixels is 1, 2, 3, 4, and 5, the certified accuracies for our method are respectively 0.746,

0.718, 0.690, 0.660, and 0.636. In contrast, the certified accuracies of Chiang et al. are 0.400, 0.369, 0.342, 0.312, 0.308. As the results show, our method is better than Chiang et al. (2020). We note that our certified robustness guarantee is probabilistic while Chiang et al. (2020) can give a deterministic certified robustness guarantee.

