# OpenReview forum: "Almost Tight L0-norm Certified Robustness of Top-k Predictions against Adversarial Perturbations"
_ICLR.cc/2022/Conference — ICLR 2022 Poster_

### Official Review · Reviewer_PF8K · 2021-10-28

**Correctness:** 3
**Technical Novelty And Significance:** 3
**Empirical Novelty And Significance:** 3
**Recommendation:** 5
**Confidence:** 3

**Main Review:**

In total, this is a good work both in the theoretical perspective and the experimental perspective. The authors provide a solid proof on the l0-norm certified robustness guarantee for top-k predictions, which is different from previous works of Levine & Feizi (2019) and Jia et al. (2020). The detailed experiments show its consistent improvement in certified top-k accuracy on CIFAR-10 and ImageNet and the ablation study provides a view on how these hyperparameters affect the performance.

**Summary Of The Paper:**

This paper provides an almost tight l0-norm certified robustness guarantee for top-k predictions against adversarial perturbations, which extends certified radius of the top-1 prediction from Levine & Feizi (2019) to that of the top-k predictions, and the l2-norm certified radius from Jia et al. (2020) to the l0-norm certified radius. The experiments on CIFAR10 and ImageNet show that the proposed method substantially outperforms state-of-the-art for top-k predictions.

**Summary Of The Review:**

The author derives an almost tight l0-norm certified robustness guarantee of top-k predictions against adversarial perturbations for randomized ablation. The corresponding theoretical analysis show both the l0-norm certified radius and the tightness. The empirical results demonstrate that the l0-norm certified robustness is substantially better than those transformed from l2-norm certified robustness.

---

> ### Author Response · Authors · 2021-11-19
> **Response to Reviewer PF8K**
>
> Thanks for the review and insightful comments.

---

### Official Review · Reviewer_5Exk · 2021-11-02

**Correctness:** 3
**Technical Novelty And Significance:** 3
**Empirical Novelty And Significance:** 3
**Recommendation:** 6
**Confidence:** 3

**Main Review:**

I appreciate the theoretical analysis and the impressive empirical results in the paper.
Concerns and questions:
* The "Deriving the certified radius for the smoothed classifier" sections is rather difficult to understand. I recommend the authors to reorganize this section and define the notations separately in a clearer way.
* The paper states that for $k=1$ the certified radius is tight. But this statement is only true if there is no assumption on the base classifier is made, and this is not true for a given base classifier, therefore this statement may be overclaimed.
* In Table 1&2, the comparisons of Top-1 prediction for 'Our methods' are missing.
* I recommend the author to compare with another L0 verification methods like https://openreview.net/pdf?id=HyeaSkrYPH.

**Summary Of The Paper:**

The paper proposed a tighter L0 verification method for Top-K predictions of classifiers based on randomized smoothing.

**Summary Of The Review:**

I appreciate the theoretical analysis and the impressive empirical results in the paper but the written is not clear. I also have some questions on the experiments.

---

> ### Author Response · Authors · 2021-11-19
> **Response to Reviewer 5Exk**
>
> Thanks for the review and insightful comments.
>
> 1. We have revised this part as suggested. Please refer to our revised paper for details.
>
> 2. Thanks for pointing this out. We have clarified that the certified radius is (almost) tight is true when no assumption is made on the base classifier.
>
> 3. Sorry for the confusion. Theoretically, our method is the same or better than Levine & Feizi + SimuEM. We found that our method has the same certified accuracy as Levine & Feizi + SimuEM on CIFAR10 and ImageNet datasets. So we didn’t show the results for our method in the tables. We have added them to Table 1 and 2 in our revised paper.
>
> 4. We note that the method in Chiang et al. (2020) is only applicable for top-$1$ prediction. We compare with the method on the CIFAR10 dataset for top-$1$ predictions using the publicly available code. The results are as follows. When the number of perturbed pixels is  1, 2, 3, 4, and 5, the certified accuracies for our method are respectively 0.746, 0.718, 0.690, 0.660, and 0.636. In contrast, the certified accuracies of Chiang et al. are 0.400, 0.369, 0.342, 0.312, 0.308. As the results show, our method is better than Chiang et al. (2020). We note that our certified robustness guarantee is probabilistic while Chiang et al. (2020) can give a deterministic certified robustness guarantee. We have cited the paper in related work and have included experimental results in Section D in Appendix.

---

### Official Review · Reviewer_NNyi · 2021-11-03

**Correctness:** 4
**Technical Novelty And Significance:** 3
**Empirical Novelty And Significance:** 3
**Recommendation:** 6
**Confidence:** 4

**Main Review:**

[Strength]
1.	The writing is concise and easy to understand in general.
2.	The paper makes nontrivial theoretical progress. Also, the authors provided enough explanation about how this progress can be distinguished from other existing works, so the contribution looks to be unique.

[Weakness]
1.	I don’t see any severe weakness (to reject this paper) in this work.

[Comments]
1.	I enjoyed reading this paper, and I think that the approach is “kind of good”. For the following reasons, I’d not put a score of 8 for this paper.
A.	The suggested proof approach is not easily generalizable, so it would be hard to say that this is a major discovery in the theory of certified robustness. (If possible, some discussion on the possible uses of the proof strategy would make the paper better.)
B.	I might have missed some potential weaknesses that I should have found. For this comment, I want to discuss with other reviewers during the review process.
2.	I can understand how the authors compared their method ($\ell_0$-certified robustness) to existing counterparts (some of them are $\ell_2$-certified robustness) and I can see there are not that many possible ways to compare them. However, I’m still not convinced whether it is a fair comparison or not, and I’m not even sure whether we can directly compare those methods or not. Also, in my personal opinion, there could be better experiment questions than just showing “our method is excelling other methods”.


**Summary Of The Paper:**

The paper provides both theoretical progress and experimental results on certified robustness of top-$k$ predictions. Specifically, on the theory side, the paper shows that the randomized ablation (Levin & Feizi) has $\ell_0$-norm certified radius of top-$k$ predictions. Also, the paper proves that the certified radius is tight for k = 1 and almost tight for k > 1. (The certified radius cannot be larger than “the derived radius plus 1”.) On the experiment side, the paper compares the proposed method to existing competitors and explored the impact of the related parameters.

**Summary Of The Review:**

I vote for accepting. First, the paper proved a nontrivial result about the certified robustness of top-$k$ predictions. Also, the paper provided enough justifications for its unique contribution to the defenses using randomized smoothing & randomized ablation type, so I believe this paper can be distinguished from other existing works. While I believe that the work itself is publishable, I’m a little bit reluctant to give a score of 8 for this paper yet due to the reasons I provided in Comments.

---

> ### Author Response · Authors · 2021-11-19
> **Response to Reviewer NNyi**
>
> Thanks for the review and insightful comments.
>
> 1. The key challenge in proving the tightness of certified robustness guarantee is that we are not able to find regions in the discrete space that satisfy certain conditions. The reason is that we cannot arbitrarily divide the discrete space into different regions. To address the challenge, we propose to relax the conditions when finding the regions to prove that the certified robustness guarantee is almost tight. The idea in our proof is very general and we hope our proof can inspire future research in proving the (almost) tightness for $\ell_0$-norm certified robustness guarantee. We have added the discussion to the end of the proof of our Theorem 2 in the Appendix.
>
> 2. We compare with the methods that are designed for $\ell_2$-norm certified robustness because we aim to show that it is suboptimal to derive $\ell_0$-norm certified robustness for top-$k$ predictions by leveraging the relationship between $\ell_2$-norm and $\ell_0$-norm. We have clarified this in our revised paper.

---

### Official Review · Reviewer_AjCc · 2021-11-03

**Correctness:** 3
**Technical Novelty And Significance:** 3
**Empirical Novelty And Significance:** 2
**Recommendation:** 6
**Confidence:** 2

**Details Of Ethics Concerns:**

None for this paper.

**Main Review:**

There are several key pros of the paper that contribute to my favorable opinion. In particular, the authors do a good job of highlighting recent and related works and ensuring that readers understand the contribution they are making relative to those works that it builds on. Further, the advances that the authors suggest in terms of improving tightness are stated in an intuitive way which allows for greater appreciation of the relatively simple modification they make. Finally, I think the introduction of top $k$ predictions rather than top 1 prediction is an interesting contribution which extends the applicability of randomized ablation (i.e. $\ell_0$ smoothing).

The primary cons of this paper are in its potentially incremental impact. I think the clear exposition of prior works may make this paper's contributions seem simple or incremental, but I think ultimately that these are important developments/extensions of prior work that constitutes a strong enough contribution for acceptance.

The second primary con (and perhaps the more noteworthy one) is the applications that this method is tested on. It is rare for the $\ell_0$ metric to be tested on images in practice. However, for graph neural networks and 3D deep learning this is a primary threat model. I think it would greatly strengthen the paper if such networks were certified and would add yet another point of contribution for this paper.




**Summary Of The Paper:**

In this paper the authors extend the tightness guarantees of other works on randomized smoothing to the $\ell_0$ case and also tighten the guarantees given by previous works by noting the discrete nature of the predictive distribution induced by randomized ablation. In addition, they extend previous $\ell_0$ methods to certify top $k$  predictions rather than just top $1$ predictions which is typically the case for smoothing.

**Summary Of The Review:**

Though one might see the contributions of this paper as incremental over other works in smoothing, I think these are valuable and important developments in the field of smoothing which can enable further applications to gain guarantees via smoothing.

---

> ### Author Response · Authors · 2021-11-19
> **Response to Reviewer AjCc**
>
> Thanks for the review and insightful comments.
>
> Thank you for your suggestions. We conduct experiments for 3D deep learning. In particular, we evaluate our methods on the ModelNet40 benchmark dataset and use PointNet as the model. We set $e=16$, $n=10,000$, and $\alpha$=0.001. When the number of modified points is 10, 20, 30, 40, and 50, the certified accuracies for top-$1$ prediction are 0.779, 0.743, 0.700, 0.649, and 0.570; and the certified accuracies for top-$3$ prediction are 0.901, 0.867, 0.829, 0.803, and 0.775. We have added the discussion to Section C in Appendix.

---

### Decision · Program_Chairs · 2022-01-20

**Decision:**

Accept (Poster)

**Comment:**

Thank you for your submission to ICLR.  The reviewers ultimately have mixed opinions on this paper, but reading in a bit more depth I don't feel that the critical comments raised by the sole negative reviewer really raise valid points.  Specifically, the fact that this reviewer directly asks e.g. for comparisons to Levine and Feiz 2019, when the paper (before its revisions) contains an entire section devoted to exactly this comparison, strikes me as not sufficient for a thorough review.

However, while I'm thus going to recommend the paper for acceptance (it does present a notable, if somewhat minor, advance upon the state of the art in randomized smoothing), I also feel the paper is generally rather borderline for more straightforward reasons.  Specifically, given the _very_ narrow focus of the proposed improvements (improvements to the bounds of randomized smoothing, for L0 perturbations, for Top-k accuracy), I ultimately don't think the paper presents that significant an advance in the field.  The paper could go other way, thought definitely not doing so due to the issues that the sole critical reviewer takes.